# Heterogeneous Autonomous Robotic System in Viticulture and Mariculture: Vehicles Development and Systems Integration [note 1]

**DOI:** 10.3390/s22082961

**Published:** 2022-04-12

**Authors:** Nadir Kapetanović, Jurica Goričanec, Ivo Vatavuk, Ivan Hrabar, Dario Stuhne, Goran Vasiljević, Zdenko Kovačić, Nikola Mišković, Nenad Antolović, Marina Anić, Bernard Kozina

**Affiliations:** 1Faculty of Electrical Engineering and Computing, University of Zagreb, Unska 3, 10000 Zagreb, Croatia; jurica.goricanec@fer.hr (J.G.); ivo.vatavuk@fer.hr (I.V.); ivan.hrabar@fer.hr (I.H.); dario.stuhne@fer.hr (D.S.); goran.vasiljevic@fer.hr (G.V.); zdenko.kovacic@fer.hr (Z.K.); nikola.miskovic@fer.hr (N.M.); 2Institute for Marine and Coastal Research, University of Dubrovnik, Kneza Damjana Jude 12, 20000 Dubrovnik, Croatia; nenad.antolovic@unidu.hr; 3Faculty of Agriculture, University of Zagreb, Svetošimunska cesta 25, 10000 Zagreb, Croatia; mseparovic@agr.hr (M.A.); bkozina@agr.hr (B.K.)

**Keywords:** viticulture, mariculture, heterogeneous robotic system, all-terrain mobile manipulator, light autonomous aerial robot, autonomous surface vehicle, remotely operated vehicle, landing platform, tether management system, acoustical underwater localization

## Abstract

There are activities in viticulture and mariculture that require extreme physical endurance from human workers, making them prime candidates for automation and robotization. This paper presents a novel, practical, heterogeneous, autonomous robotic system divided into two main parts, each dealing with respective scenarios in viticulture and mariculture. The robotic components and the subsystems that enable collaboration were developed as part of the ongoing HEKTOR project, and each specific scenario is presented. In viticulture, this includes vineyard surveillance, spraying and suckering with an all-terrain mobile manipulator (ATMM) and a lightweight autonomous aerial robot (LAAR) that can be used in very steep vineyards where other mechanization fails. In mariculture, scenarios include coordinated aerial and subsurface monitoring of fish net pens using the LAAR, an autonomous surface vehicle (ASV), and a remotely operated underwater vehicle (ROV). All robotic components communicate and coordinate their actions through the Robot Operating System (ROS). Field tests demonstrate the great capabilities of the HEKTOR system for the fully autonomous execution of very strenuous and hazardous work in viticulture and mariculture, while meeting the necessary conditions for the required quality and quantity of the work performed.

## 1. Introduction

The automation and robotization of technological processes has greatly increased and improved food production. The invention of heavy machinery such as tractors and combines has reduced the need for human labor in most areas of agriculture, but some segments, such as viticulture, are still very labor-intensive. In viticulture, most vineyards are located in hilly terrain, which increases the amount of labor required for successful growth. Many new vineyards in Mediterranean countries are now established on very steep terrain, where tasks such as protective vine spraying and manual suckering require extreme physical strength and endurance from workers, making these tasks particularly interesting for the introduction of robotic solutions. With such robotic systems, characterized by autonomy, adaptability and consistency with the objectives of organic farming, it is possible to replace workers and relieve them of strenuous work.

Agriculture has been part of human society for thousands of years [1]. Aquaculture began its exponential growth only in the 20th century [2], together with its part related to salt water called mariculture. The tasks of monitoring net pens for fish farming require the long-term work of divers in all, even the most difficult, weather conditions. In addition, the regulations of environmental protection and constant monitoring of the condition around and under the net pens in mariculture necessarily require the measurement of all relevant parameters and the collection of samples from the bottom (sediment) and in the water column.

The solution of these problems in modern viticulture and mariculture is sought in the project HEKTOR—Heterogeneous Autonomous Robotic System in Viticulture and Mariculture [3,4]. The main goal of the HEKTOR project is to find a systematic solution for the coordination of intelligent heterogeneous robots/vehicles (marine, terrestrial and aerial) capable of cooperating autonomously with each other in the open, unstructured space/water environment. HEKTOR is a robotic system consisting of four different types of robots that communicate and cooperate via an ROS-based middleware. They are a ground robot in the form of an all-terrain mobile manipulator (ATMM) for vineyard work, a lightweight autonomous aerial robot (LAAR) for both viticulture and mariculture scenarios and a remotely operated underwater vehicle (ROV) and an autonomous surface vehicle (ASV) for maritime work. The system overview of HEKTOR vehicles is shown in Figure 1.

The main contributions of this work are (1) the heterogeneity of the proposed robotic system in terms of including aerial, ground and marine vehicles—to the best of the authors’ knowledge, no such system exists in the literature, as explained in more detail in the Section 2 and Section 3. Therefore, the concept of such a system, together with the design of the vehicles and the integration of the subsystems, is the main novelty. The second is (2) the heterogeneity of applications for the proposed system, i.e., the same system can be used in both viticulture and mariculture, with the LAAR being the link; the third is (3) the use of an autonomous cooperative multi-robot system in unstructured environments such as steep karst vineyards and fish farms to solve the problem of tedious and often dangerous manual labor.

### Organization of the Paper

This article is structured as follows. Section 2 describes the overview of viticulture scenarios in which autonomous vehicles could be used and the related literature. Section 3 provides a similar overview, but for mariculture scenarios. Section 4 describes the robots of the HEKTOR system. The development of the all-terrain mobile manipulator (ATMM) is described in Section 4.1. The light autonomous aerial robot (LAAR) is presented in Section 4.2. Section 4.3 describes the Blueye Pro remotely operated vehicle (ROV) and its integration into the ROS framework. The development of the Korkyra autonomous surface vehicle (ASV) is described in Section 4.4. Section 5 describes the HEKTOR subsystems for robot collaboration, namely the landing platform (LP) (Section 5.1), the underwater acoustic localization system (UWGPS) (Section 5.2) and the tether management system (TMS) (Section 5.3). An overview of the Robot Operating System (ROS) framework and its use as a tool for the autonomy of individual vehicles as well as collaboration between all vehicles is described in Section 5.4. Finally, Section 6 concludes the paper with some directions for future work.

## 2. Viticulture Scenarios

Viticulture includes a number of activities required for cultivating a successful vineyard, such as vineyard surveillance, weed management, pruning, suckering, spraying, harvesting and others. Of these activities, the HEKTOR project focuses on vineyard surveillance, spraying and suckering (Figure 1), as they are all extremely difficult when working on large slopes is considered, typical of new vineyards on the Adriatic coast and islands.

This section presents various applications of the HEKTOR system in viticulture, together with a literature overview for vineyard surveillance, spraying and suckering. All the approaches found in the literature are described in Table 1 with respect to the tasks, vehicles and environment structure.

### 2.1. Vineyard Surveillance

The regular monitoring of the vineyard is an important aspect of viticulture. In this way, the vineyard manager collects information about the current condition of the vines, the occurrence of diseases and the yield of grapes. Therefore, researchers are primarily focused on visual recognition of grapes for the purpose of yield estimation, as well as early detection of grapevine diseases. In [5], the authors estimate vineyard yields based on video recordings made by a manually controlled vineyard utility vehicle. Yield estimation by an autonomous ground robot is presented in [6]. Autonomous ground robot navigation for monitoring vineyards with steep slopes is presented in [7]. Various combinations of UAVs equipped with thermal imaging, multispectral and high-resolution cameras for monitoring vineyards are presented in [8,9,10,11,12,13].

In HEKTOR, the LAAR will perform autonomous flights and create different variants of the vineyard map depending on the sensors used. It is also necessary to recognize the structure of the vineyard (main and secondary roads, plantation rows). The goal is to provide the system operator with information about the condition of the vineyard, but also to create a global map of the vineyard, which is needed for the navigation of the mobile manipulator. One of the main advantages of HEKTOR is the monitoring coordination of LAAR and ATMM, which will contribute to the accuracy of information about the condition of the vineyard.

The creation of a point cloud and a digital elevation model (DEM) of the surveyed terrain and the vineyard can be done by acquiring and processing the data generated by the LiDAR sensor or by the photogrammetry process. The input data for the photogrammetry procedure are a set of georeferenced images acquired by the LAAR. The images are acquired during the LAAR’s flight mission in the vineyard (Figure 2)—the LAAR performs a lawnmower trajectory over the area of interest, during which high-resolution images are acquired. Each image must contain approximately 60% overlap information with adjacent images, as well as the Global Navigation Satellite System (GNSS) coordinates of the position at which it was taken. The photogrammetry process generates keypoints on each image based on the features of the photograph and matches them with the same keypoints on other input images. The result of this process is a generated point cloud and a 3D model of the surveyed area (Figure 2). The generated point cloud and 3D model contain information about the terrain, as well as models of the vines and other man-made infrastructure in the area. The digital elevation model of the vineyard is created by restructuring the data from the point cloud into a 2D array with elevation information. DEM can then be used as a basemap for planning LAAR flight missions, where the LAAR can maintain a constant altitude with respect to the terrain and not just the launch point. Such a description of the vineyard and the collected input data led to a methodology that allows the estimation of energy consumption before the pure deployment of the described robotic system [20].

### 2.2. Vineyard Spraying

Conventional vine protection requires a lot of protective chemicals because the vine is highly exposed to many pests (e.g., red spider, grape berry moth) and diseases (e.g., downy mildew, powdery mildew, gray mold and black rot disease). Conventional protection requires many sprays with various agents such as insecticides, fungicides and acaricides. The result of spraying is the introduction of a significant amount of chemicals into the immediate environment of the vineyard. Nowadays, the chemicals are applied with blower-like machines that create a cloud of chemical droplets. However, many of the droplets do not land on the leaves. A significant portion of the chemicals end up in the soil or in the air [21,22]. These chemicals can have serious consequences for the health of farmers [23]. The use of robots for vineyard spraying opens up a number of opportunities to reduce the amount of chemicals introduced into the environment by using artificial intelligence to optimize the chemicals used to the minimum necessary.

Some research efforts have already been put into the development of robots for spraying tasks in agriculture [14,15,16,17]. In [14], Berenstein et al. present an autonomous vehicle for spraying vineyards equipped with multiple spray nozzles placed at fixed positions at different heights. They use a foliage detection algorithm and decide which nozzles should be active at a given moment based on its results. Oberti et al. [16,17] use multispectral cameras to detect disease-prone areas on grapevine leaves and spray these areas with a precision nozzle mounted on the robotic manipulator.

One of the main goals of the HEKTOR project is to create a robotic system that successfully performs the task of spraying. At the same time, achieving this goal would make it possible to set new goals for optimizing spraying. In the HEKTOR project, spraying vineyards involves the use of an ATMM equipped with a spraying system to perform a protective spraying task. The ATMM must autonomously move through the vineyard rows at an adjustable speed using a map and information collected by the LAAR and its own sensors. It is also necessary to achieve the best possible spraying angle. In the scenario studied, the ATMM travels in a straight line along a vineyard row while the robotic arm controls the relative pose of a spray nozzle with respect to the mobile base (Figure 3). A depth camera is used to generate a lawnmower spray trajectory that corresponds to the general shape of the foliage along the row. In this way, we attempt to minimize the waste generated by spraying while maintaining satisfactory coverage. More details about the first experiments and effectiveness of the controlled spraying approach can be found in [24].

### 2.3. Suckering

Suckering is the removal of undesired shoots that originate from the trunk in the spring. These shoots consume energy, water and nutrients from the developing fruit. This procedure is usually done by hand in a stooped position, which can negatively affect the health of farmers [25], but there are several alternatives. The commercial suckering machine shown in [26] uses a mechanical system to remove the shoots, but in the process, the entire trunk is hit, which in turn can damage it. In [18], the authors present a method for detecting shoots for targeted spraying using optical sensors. In [19], the authors suggest using flaming for suckering.

Suckering is one of the HEKTOR system goals that requires the mobile manipulator to move autonomously through the vineyard, locate the vine, position itself next to it and do the job. It is also necessary to autonomously remove undesirable shoots without damaging the trunk. Navigation in the vineyard by the ATMM is carried out based on the map generated by the LAAR, which is then used together with measurements from its sensors for precise navigation and an accurate approach to each vine.

The task of suckering is significantly complicated by the presence of wires near the vine. Accordingly, this task can be approached in a number of ways, from the simplest, which uses dedicated rubber threshing mechanisms, to the most complex, which uses depth cameras and compliant end effectors and detects the shape of the trunk, determines the position of the wire and controls the movements of the suckering tool. The most complex possible approach would be to get near the trunk, create an accurate 3D model and then use that for the precise autonomous removal of undesirable shoots. More complex approaches would require significant computational effort that would slow the execution of the task to a level that could become unacceptable for the use of the robotic system. Regardless of which approach to solving this task would be faster and more efficient, the development of suckering solutions in the HEKTOR projRefect goes both ways. In Figure 4, the conceptual design of the tool with a height-adjustable rubber threshing mechanism is shown on the left, and the compliant, hydraulically actuated gripper is shown on the right.

## 3. Mariculture Scenarios

The HEKTOR project involves the operation of the autonomous surveillance of fish net pens using a heterogeneous robotic system (Figure 1). This process involves three types of robots, LAAR, ROV and ASV, which are described in Section 4.2, Section 4.3 and Section 4.4.

In the pen culture technique, the net pen may completely enclose the culture volume (immersed pens), but more often, they enclose it on five sides, while the surface of the net pen is not enclosed because it is, in itself, a natural barrier to the organisms being cultured (surface pens). Once the net pens are in the sea, they must be regularly cleaned and repaired or replaced to prevent excessive growth of organisms.

When fouling occurs, the net pens are first colonized by algae, which are favored by ammonia emissions from the farmed fish. Thereafter, various invertebrates may colonize the net, depending on the time of year at which the net pen is in the sea. Essentially, the higher the seawater temperature, the higher the fouling rate. Net pen overgrowth poses a threat to farmed fish because it (i) impedes the flow of seawater through the net, which also reduces the amount of altered water in the pen; (ii) aggravates the net pen, increasing the possibility of damage and fish escaping from the pen; (iii) increases the weight of the pen, which increases friction on the mooring system; (iv) creates additional biomass of organisms that may contribute to excessive oxygen consumption in the pen.

Today, net pens are cleaned using commercially available machines for underwater net cleaning (Figure 5a). Regardless of the development of underwater net washing technology, nets must be pulled from the sea and thoroughly washed at regular intervals. Conventional methods of monitoring and maintaining marine fish farms are currently conducted exclusively manually (farm staff and/or hired divers inspect the condition of net pens above and below sea level, net fouling, damage to nets, etc.) or using aerial robots and remotely operated underwater vehicles (ROVs), which simplify and speed up the monitoring process but do not automate or reduce the cost of paying professional vehicle operators or training farm staff to work with the vehicles.

This section presents various applications of the HEKTOR system in mariculture, together with a literature overview for net pen inspection and fish stock biomass estimation. An overview of all approaches found in the literature related to HEKTOR is given in Table 2 with respect to the tasks, vehicles and environment structure. Looking at Table 2, it is important to note that out of the 17 chosen net pen inspection and biomass estimation tasks in mariculture, most use only one vehicle (an LAAR or an ROV) to perform. Only one approach uses an ROV connected to an ASV to perform the inspection task. Net inspection-related approaches either use datasets recorded at fisheries for benchmarking algorithms or they deploy actual vehicle(s) at net pens. Thus, most of them are regarded as working in unstructured environments. Almost none of the biomass estimation methods shown in Table 2 use any robots. Only the last listed method, published by the authors, proposes a method that uses at least an LAAR and an ROV, with the option to control the ROV by an ASV. Moreover, none of these methods mention using robots in an unstructured environment for online fish counting and biomass estimation since most propose either fixed structures for stereo vision systems or develop image processing algorithms based on real-world datasets.

### 3.1. Net Pen Inspection

There are several existing solutions that use aerial vehicles for the inspection and maintenance of net pens. In [27,28], the authors characterize the movement of spatial pellet distribution, determining spatial feed distribution by a UAV. The authors in [29] present an aquaculture pen detection method based on a UAV-mounted camera, while the authors in [30] present image processing for aerial imagery used in lake resource exploitation, aquaculture and coastal communities. In [31], the authors propose a method for feeding fish in aquaculture using an aerial vehicle.

Underwater vehicles are used in various applications for inspection and monitoring, and some of them are related to aquaculture. In [32], a solution for navigating a net pen based on laser camera triangulation is presented, while in [33], the authors present an autonomous underwater vehicle for the automatic inspection of net pens. A similar system is presented in [34], where the authors present an integrated ROV solution for the underwater inspection of net pens in fish farms. A system consisting of an unmanned surface platform and an ROV for marine farm inspection in Norwegian aquaculture is presented in [35].

The HEKTOR project aims to develop solutions for the aerial inspection of net pens using LAAR and for underwater inspection using a combination of ROV and ASV. The end results of the project will be the coordination of LAAR, ROV and ASV for the inspection of net pens from the air, from the sea surface and from below the sea surface. The data from all three robots will be used for the final assessment of the net pens based on the data fusion of the individually collected information.

Upon approaching the net pen, both LAAR and ROV are docked to the surface platform (ASV). Upon arrival, the LAAR lifts off and the ROV separates from the ASV. For autonomous aerial inspection, the LAAR must be positioned autonomously above the net pen and collect video data on the structure of the pen and the activity of fish at the sea surface. The LAAR must be able to detect the net pen in the video image (Figure 5b) and position itself over the pen using image processing. It must also be able to autonomously take off and land on an autonomous surface platform near the net pen.

For the underwater inspection of net pens, the ROV must be able to navigate around the entire submerged portion of the net pen and capture video (Figure 5c). Localization of the ROV with respect to the net pen will be accomplished by an acoustic localization system. For this purpose, there should be cooperation between the ROV and the surface platform (ASV), i.e., the surface platform must constantly monitor the position of the ROV to reduce the number of cables connecting it and the surface platform (ASV) in the sea, thus reducing the disturbance caused by the sea currents.

Photographs and video taken by the ROV can also be used to measure fouling on the net pen to estimate both the composition and amount of fouling flora/fauna. Based on the known mesh size and amount of fouling, the restriction on the flow of seawater through the net can be estimated and thus a recommendation made for the replacement or cleaning of the net pens. The parameters that can be estimated from the photographic material are: total fouling and approximate composition and restriction of surface area for water exchange through the net.

By placing a measurement device (multi-parameter probe/logger) on the ROV, it is possible to measure the physicochemical parameters of the seawater both in real time and in the microenvironment of the net pen. The data obtained from the measurement could be used to estimate the amount of seawater flowing through the pen. The parameters that this measuring device would measure are the following: pH, water redox potential, dissolved oxygen, water turbidity (cloudiness), temperature, salinity and dissolved nitrates/nitrites.

### 3.2. Fish Population Modeling

The problem of counting fish and estimating their size has been studied by many research groups. The problem of fish counting on a smaller scale is presented in [36,37,38,39,40]. The estimation of fish weight and its classification is studied in [41,42,43]. All these approaches are based only on information collected below the sea surface.

In the HEKTOR project, information collected both below the sea surface and from the air is analyzed for fish population modeling. Analysis of photo and video footage allows non-invasive sampling of fish populations at high frequency, even in real time. Using software to analyze photo and video footage captured by a high-resolution camera on the ROV, the population density of farmed fish and the average size of individuals can be measured. The parameters that the system can process from the photos are the size of the individuals, the number of fish in the field of view and possible damage to the fish’s body. The obtained measurement data are processed statistically and a model of the growth of fish in the farm over time is created. During fish feeding, when large numbers of fish move closer to the surface, the information collected by the LAAR can be used to estimate the number of fish and the average size of individuals. As the authors of this paper suggest in [44], a reasonably good estimation of the biomass can be achieved if it is assumed that the fish follow normal distribution in the horizontal plane at the surface during feeding. Coupled with the assumed exponential fish distribution that decreases with depth, the total number of fish and the resulting biomass in the net pen can be estimated.

## 4. Robots of the HEKTOR System

The HEKTOR system consists of four different types of robots used for the scenarios described in Section 2 and Section 3. Table 3 shows some of the requirements that the HEKTOR robot system must meet and that ultimately influenced the design of the system.

In this section, each robot is presented with its specifications and capabilities. Table 4 shows the main specifications and features of each robot.

### 4.1. All-Terrain Mobile Manipulator (ATMM)

In the HEKTOR project, the purpose of the ATMM is to autonomously move through the vineyard and perform vineyard operations that include spraying and suckering [45].

In addition to meeting the requirements listed in Table 3, the main objective of the presented ATMM design was to ensure safe robot movements in rough terrain and to safely perform various tasks under rather adverse operating conditions in very steep and rough terrain. A primary reason for using tracks instead of wheels was to operate on slopes of up to 60%. Other design goals were easy maintenance, high energy efficiency, multifunctionality and flexible expandability of the robot.

To achieve all these goals, a modular design concept of a mobile platform driven by four independently controlled flippers/tracks is proposed. As shown in Figure 6, the ATMM structure consists of three main modules that are fully separable:ia chassis with four flippers/tracks,iian electrical compartment andiiia robot arm carrying various robot tools (e.g., spray nozzle, soft gripper).

The body of the mobile platform consists of an aluminum skeleton, a central steel cylinder that serves as housing for the main battery stack and two identical steel tanks at the front and back of the robot to store a protective spray liquid. Two side tanks provide space for attaching flippers/tracks. Access to the main battery is very easy, and when the battery is depleted, it can be replaced with a charged one in a very short time. In this way, extended downtime due to battery recharging can be avoided and reduced to minutes. In addition, the mobile robot base includes a mechanical assembly for mounting a robot manipulator for spraying, suckering, etc.

An electrical compartment contains control, sensor and wireless communication modules, including a gimbal interface for operation with standard, multispectral and depth (RGB-D) cameras. In addition, the robot is equipped with the appropriate flow/pressure control components (pump, control valves, flow meter, pressure gage) required for controlled spraying, which is a prerequisite for minimizing liquid waste and associated environmental impact. Due to the very stable construction of the robot, it can carry an additional container for the deposition and transport of the protective liquid.

The goal of the robot arm is to carry various tools while providing the necessary movement. Two types of tools are used—a soft, force-controlled gripper used for sensitive suckering tasks (see Figure 4) and a nozzle designed for the selective spraying of a vine foliage (see Figure 6). In the case of protective spraying, the positioning of the tool is performed by controlling four flippers/tracks and a robotic arm. In addition, the use of an articulated robot arm allows the spraying of a vine canopy from different approach angles. As can be seen in Table 3, an average planting height of 1.5 m dictates the dimensions of a robot arm that must be mounted on the mobile robot base. The choice of the Kinova Gen3 robotic arm was advantageous not only because of its sufficient vertical reach, but also because it is a lightweight robotic arm with torque-controlled joints that allow control of the contact force exerted while performing the suckering task.

Some of the requirements for the effectiveness of the HEKTOR system mentioned in Table 3, such as the average ATMM speed (0.7 m/s) and suckering rate of the robot arm (20 vines per hour), have yet to be confirmed in practice.

The advantages of such a robot design are as follows:-Easier control of the robot center of mass (CoM), since the flippers are relatively large compared to the robot body (each flipper contains its servo drive and control electronics);-More efficient overcoming of obstacles due to better distribution of the force to the flipper tracks, which contribute more than others to the robot’s movement upon contact with the ground;-Easier maintenance of the robot as each flipper is an independent module that can directly replace any other flipper module;-The raised center section of the robot skeleton prevents the robot from frequently colliding with or getting caught on the outer (top) edges of obstacles.

The size of the robot makes it easy to traverse corridors between vineyard rows. In addition, the weight of the robot is light enough to allow easy storage and handling of the robot.

In the robot design presented, each flipper/track is driven by a BLDC servo module that is fully integrated into the body of the flipper/track.

The flippers have some fixed operating positions (which can be manually selected before starting the robot) to facilitate transportation and to better adjust the robot’s center of gravity when working in steep vineyards.

ATMM control has been implemented by using the Intel NUC 10 mini PC. This device supports Ethernet and CANOpen communication standards. The operator drives the mobile manipulator in the initial working position using a joystick and through a graphical user interface (GUI) sets the desired parameters of the vineyard treatment. When the operator starts the program with the desired settings, the positioning algorithms enable:continuous step-by-step passing over a vine canopy surface (in case of spraying);coordinated control of the movement of the mobile base and the robot arm;navigation that is always georeferenced and accurate with respect to the vine undergoing a particular treatment;force-controlled movement of a soft gripper on the vine (in case of suckering).

As shown in Figure 6, an all-terrain mobile manipulator is equipped with a number of sensors. LiDAR provides the control system with a three-dimensional point cloud of the environment, which is used in localization and mapping algorithms. The high-precision GNSS RTK module is also used for this purpose, along with an Inertial Measurement Unit (IMU). For vineyard spraying, a spray nozzle is mounted as an end effector of the robot arm, allowing precise control of the spray angle with respect to the mobile base. Pressure and flow feedback are used to control the amount of spray applied to the plant.

All software development is based on the use of ROS, which is described in Section 5.4.

### 4.2. Light Autonomous Aerial Robot (LAAR)

The Light Autonomous Aerial Robot (LAAR) (see Figure 7) used in the HEKTOR project is a medium-sized carbon-based quadcopter powered by two LiPo batteries (Figure 2). The dimensions of the LAAR are 1200×1200×450 mm and a mass of 8 kg to 10 kg, depending on the sensor equipment installed on the UAV. The flight time of the LAAR is around 30 min, with the possibility of manual flight mode and performing autonomous flight missions. The maximum wind speed for safe operation of the LAAR is 15 m/s. The propulsion system consists of four T-motor P60 KV170 motors with foldable 22.4×8.0 propellers, capable of generating a maximum thrust of 68 N. The LAAR flight control system consists of the primary Pixhawk autopilot module and the Intel NUC on-board computer used to implement complex control algorithms and real-time data processing from sensors.

The primary sensor for LAAR localization in the HEKTOR project is a GNSS positioning sensor with the ability to use Real-Time Kinematics (RTK) correction for higher accuracy. In addition to the GNSS sensor, the LAAR is equipped with a camera used for precise localization over specific markers in the environment. Visual servo control is primarily used when landing the LAAR on a platform mounted on the ASV.

Depending on the task that LAAR performs during the flight mission, it can be equipped with different payloads. The sensors used in the HEKTOR project are as follows:Velodyne Puck LiDAR is a 3D localization sensor with 16 channels, 360° horizontal and 30° vertical field of view. It can be used to create point cloud maps of the terrain or to localize LAAR in a GNSS denied environment.Sony RX100 high-resolution RGB camera with 1𠄳 type CMOS sensor, 3× optical zoom and the ability to record 4K video at 25 or 50 fps.Flir Duo Pro R camera that combines high-resolution radiometric thermal imaging and 4K visual sensors. During vineyard survey flights, the thermal imaging camera can be used to detect areas of elevated plant temperature that may indicate disease or drought.Micasense RedEdge-MX is a multispectral sensor that can pick up 10 narrow bands in the visible and near-infrared light spectrum. It can be used to calculate various indices, such as the Normalized Difference Vegetation Index (NDVI), in surveyed areas.

### 4.3. Remotely Operated Underwater Vehicle (ROV)

The Blueye Pro ROV (see Figure 8a) is manufactured by the Norwegian company Blueye Robotics. Its dimensions are 485×257×354 mm (L×W×H). It weighs 9 kg in the air, is designed for a depth of up to 300 m (with a 400 m tether) and has an autonomy of 2 h provided by its 96 Wh battery. It can be easily trimmed for different underwater environments, be it saltwater, brackish water or freshwater. The ROV has a total of four 350 W thrusters. Two rear thrusters in the horizontal plane, one in the vertical plane and a side thruster allow the vehicle to be highly maneuverable along with automatic heading and depth control modes. It can reach a top speed of up to 3 knots and operate in water currents of 2 knots maximum.

The main sensor is a HD camera with a [−30°,30°] tilt angle mechanism that operates at 25–30 fps. The Blueye PRO ROV features powerful 3300 lum lights with 90 CRI LEDs that provide well-lit images and excellent colour reproduction. Other sensors include an IMU with 3-axis gyro and 3-axis accelerometer, depth sensor, magnetometer (compass), temperature (indoor and outdoor) and an internal pressure sensor. Additional payload can be attached to the top and/or bottom. The ROV comes with a surface unit that allows the ROV to be connected and controlled via WiFi as well as via Ethernet in challenging wireless environments. For use in the HEKTOR project, the ROV is connected to the ASV Korkyra via an Ethernet cable. Thus, communication between the operator and the ROV is achieved using ASV Korkyra’s NUC main computer as a communication relay.

The Blueye Software Development Kit (SDK), based on Python3, was integrated into ROS2 to allow the operator to monitor the situation underwater while viewing the video stream in the ROS2 graphical user interface (GUI). The ROV-ROS2 GUI displays the ROV battery charge level and enables/disables the automatic depth and automatic heading diving modes. It also switches between manual mode (control via a gamepad connected to the operator’s computer) and autonomous mode (in which the NUC main computer aboard ASV Korkyra sends controls based on images recorded by the ROV). The ROV’s camera parameters (exposure, white balance, hue, bit rate, frame rate and resolution), as well as lighting, water density and boost gain, can be easily changed through this GUI, as shown in Figure 8b. All parameters can be easily changed using buttons, sliders or combo boxes that prevent any of the parameters from being set outside of their range. The code for the Blueye SDK-ROS2 interface and GUI is publicly available at [46].

### 4.4. Autonomous Surface Vehicle (ASV)

The autonomous surface vehicle named Korkyra (see Figure 9) is being developed as part of the HEKTOR project [4,47]. It was designed as a catamaran to provide better stability and hydrodynamic properties in sea states up to state two. It is made of aluminium, 2000 mm long, 1000 mm wide and has hollow hulls with a diameter of 240 mm. It has a modular design. The ASV is designed to withstand sea states up to 2 at inshore fish farms in the Croatian part of the Adriatic sea. It is a research vessel and not a market-ready product. Statistical analysis of Croatian offshore Adriatic sea wave heights presented in [48] and later extended in [49] clearly shows that the average wave height is well below 1 m. This is most probably due to the fact that this part of the Adriatic sea is very well indented, with many islands and islets breaking the waves. Most of the fish farms in Croatia are located closer to the shoreline or between islands to make weather conditions more favorable. This shows that the proposed sea state 2 is a good estimate that was used as a design input. Furthermore, if, at some point, there will be a need for withstanding a greater sea state, the catamaran’s hull can be easily made larger and more stable while leaving all other subsystems practically the same.

The upper deck consists of a carbon hull that houses all the electronics and computers that enable the autonomy of the vehicle. The catamaran’s lower deck houses IP67-rated watertight aluminium boxes that house additional batteries, motor electronics, the NORBIT iWBMSc multibeam sonar system INS and an expansion box that provides easy plug-and-play power, Ethernet and USB connectivity for each payload. The sonar, which is used in bathymetric applications of the ASV, is mounted in the front of the ASV on a bracket that can be lowered or raised as needed. The roll bar above the upper deck allows integration of maritime signal lights, surveillance camera(s), possibly even LiDAR, etc. It can be easily folded down if a landing platform or other payload is to be mounted above the carbon hull. The design of the carbon hull and the electronics that it contains is based on the aPad vehicle previously developed at the Laboratory for Underwater Systems and Technologies (LABUST) [50,51]. The catamaran is 800 mm high from the bottom of the hull to the top of the carbon hull and 1400 mm high to the top of the roll bar. It currently weighs 100 kg in the air (see Table 4). Mobility of the ASV on land and during deployment from shore is provided by two rugged wheels on the rear of the hulls and a swivel wheel on the front. When the ASV is deployed, the rear wheels can be easily lifted and fixed with pins, while the front wheel can be easily removed. The ASV’s stability and maneuverability at sea is further enhanced by 300 mm high keels at the front of each hull. On land, these keels act as front legs on which the ASV stands, and they also protect the payload mounted beneath the ASV.

The ASV Korkyra can be equipped with various payloads, such as a multibeam sonar, a remotely operated vehicle (ROV), a tether management system (TMS) for the ROV or a landing platform for the LAAR, as conceptually shown in Figure 1, on the upper and front lower parts of the ASV. It can accommodate an additional 50 kg (in the air) of payload so that the hulls are semi-submerged for the best hydrodynamic characteristics. If needed, it can be mounted with a maximum 100 kg (in the air) of payload when its hulls are fully submerged, which would give the catamaran slightly worse hydrodynamic characteristics. Four 390 W electric T200 thrusters in X configuration enable it to navigate complex marine environments at lower speeds (1–2 knots) in all directions. The thrusters are mounted on movable masts that can be adjusted in height. The thruster orientation in the horizontal plane can also be adjusted with a resolution of 45°. In addition, the 720 W Minn Kota RT 55 EM booster electric motor enables top speeds of 3–4 knots. The booster motor is mounted in the rear of the ASV on the same height-adjustable bracket as the sonar.

The total energy that the ASV can draw from its batteries is equivalent to 14 BlueRobotics 14.8 V, 18 Ah batteries, for a total of 252 Ah or 3.73 kWh. Worst-case calculations were performed to obtain a lower bound on the autonomy of the vehicle. Assuming that all subsystems of the ASV operate at maximum power all the time, the system power is approximately 1 kW. If the Minn Kota booster motor is not used at all, which is a realistic scenario in the fish net pen environment, where slow and highly precise movements are required, the ASV consumes at most around 200 W. Furthermore, this means that the autonomy of the ASV Korkyra is between 3.5 h and 20 h. This means that in mixed consumption scenarios, where the ASV needs to use a high-power motor to get from shore to fishery quickly and also perform slow inspection maneuvers in all directions, the ASV has an average autonomy of 10–11 h. This is more than enough for the use cases of the HEKTOR project.

The GNSS with inertial navigation system IMU, named Applanix SurfMaster, combined with base station corrections over the Long-Term Evolution (LTE) network, enables the ASV to localize itself globally with an accuracy of up to 10 cm, which is very important in survey and inspection missions. The autonomous behavior of the vehicle is supported by two Intel NUC mini PCs. One seventh-generation NUC is used to collect and process multibeam sonar data under Windows, while another tenth-generation NUC under Linux and ROS is used for mid- and high-level control, data processing and mission control. Communication with the vehicle is via WiFi (Ubiquity Bullet M2 and an omnidirectional antenna on both the ASV and operator side) with a peak transmission speed of 100 Mbps over 400–500 m range. An option with Ubiquity Rocket M2 and Ubiquity 120° sector antenna is also being tested to achieve a longer range and better bandwidth. Operator work and mission planning for ASV Korkyra will be facilitated by open-source, graphical user interface-based software called Neptus, developed by the Laboratório de Sistemas e Tecnologia Subaquática (LSTS) of the University of Porto. Neptus communicates with the vehicle via Intermodule Communication (IMC) messages, which are simultaneously integrated with ROS via an ROS–IMC bridge.

Remote monitoring of the catamaran via the video stream from a pan-tilt-zoom (PTZ) Hikvision IP camera (mounted on top of the roll bar; see Figure 9) ensures that the operator can respond to any risky situations that might occur at sea. The IP camera is also integrated with ROS, allowing the operator to receive visual feedback from the camera via WiFi. The ROS -Qt (RQT) GUI, which includes the video stream and PTZ control, is shown in Figure 10. In addition to the Real-Time Streaming Protocol (RTSP)-based video stream, this GUI also allows the operator to control the pan, tilt, zoom, focus and aperture of the camera lens.

## 5. HEKTOR Subsystems for Robot Collaboration

The robots described in Section 4 must perform tasks that require their networking and coordination. This is especially important when autonomous task execution is considered. This section describes several specific subsystems developed to enable coordination and collaboration among the robots of the HEKTOR system.

### 5.1. Landing Platform

As mentioned in Section 3, LAAR will conduct aerial inspections of fish net pens. This will be done in coordination with ASV Korkyra, so a landing platform (LP) will be needed for LAAR to dock with ASV. The landing procedure is divided into two phases. In the first phase, the LAAR detects the ASV and the landing platform with a recognizable “H” symbol and activates electromagnets at all four corners of its landing skis. Once the LAAR is firmly attached to the landing platform, two strong metal guides, mounted on pulleys and driven by a stepper motor close to the center of the landing platform, dock the LAAR. The LAAR then deactivates the electromagnets to avoid compass deviation.

The model of the landing platform is shown in Figure 11a. LP is 1000 mm × 1000 mm × 10 mm (L × W × H) in size. It can be easily mounted on the upper deck of the ASV Korkyra with a 870 mm × 870 mm × 200 mm (L × W × H) removable frame. The platform is made of a lightweight iron sheet that allows the LAAR’s electromagnets to be fixed to the platform during the initial phase of the landing procedure. An “H” is cut out in the center of the platform to reduce the weight of the platform, but also to serve as a marker for the LAAR during landing. The iron sheet is covered with a protective layer to prevent it from rusting due to the salty and humid marine environment. The landing platform is connected to the upper deck of the ASV with bolts and nuts so that it can be easily removed from the ASV for missions where the LAAR is not needed, as shown in Figure 11b.

LP’s control system consists of two Nema 17, 59 Ncm, 2 A bipolar stepper motors connected in parallel to the DM542T digital stepper motor driver with an output current range of ${I_{driver}^{LP}}^{out}$ = 1.0–4.2 A and an input voltage range of ${U_{driver}^{LP}}^{in}$ = 20–50 VDC. For the first prototype, a stepper motor with encoder and/or IP65 protection was not chosen because it is inherently heavier. This is a very important factor when building an ASV with so many subsystems. Therefore, a lighter stepper motor was chosen to test whether its maximum load holding torque is sufficient for this application and whether and how often it skips steps. Metal guides that open and close are connected to a lead screw on each side of the landing platform. The motor shafts are connected to the lead screws by a belt transmission system with a transmission factor of $K_{tr}=1/3$. The transmission factor of 1/3 causes the lead screws to rotate more slowly, but more importantly with a higher torque. This is especially important when the LP closes the guides. In this case, it must center the LAAR if it does not land in the center.

The control loop is closed over four IP65 rated contact switches (see Figure 11c) that detect when the LP is open or closed so that the control algorithm knows when to stop the motors. The reference rotation speed noted as rpm was lowered to 510 rpm based on empirical tests of motor torque under load performance.

A large number of test runs were performed to identify potential problems with the LP. Extensive testing was first performed in the laboratory environment with the prototype without the four contact switches that serve as feedback; see Figure 12a. These tests showed that the selected NEMA 17 motors have sufficient torque to hold the LAAR prototype landing skis in both longitudinal and lateral directions once the guides are fully closed. Outdoor testing was conducted in September 2021 at the University Campus in Zagreb, Croatia, as shown in Figure 12b. The strength of the LP structure was tested to determine if it could withstand the full mass of the LAAR. During these tests, the LAAR was manually landed at LP so that operators could get a feel for the precision that their autonomous landing algorithm must have. All of the above tests were successful. In addition, the motors were tested to see if they could drive the guides to center the LAAR and close in place, even when the LAAR lands at an angle. It has been shown that this is not possible when the LP motors are running at ${rpm}_{max}=750$ rpm, as they then do not have enough torque to push the LAAR into place. One way to solve this problem is to use a much lower rpm throughout the maneuver, but this would slow the maneuver down considerably. Another option is for the guides to travel at top speed as long as they do not touch the LAAR’s landing skis, and then slow down significantly.

### 5.2. Underwater Acoustic Localization System

Control of the ROV by the ASV requires position feedback. In the underwater environment, this means using an acoustic localization system in most cases because electromagnetic waves are highly attenuated. Since the horizontal plane dimensions of the ASV Korkyra are 2000×1000 mm, using a range-based short baseline (SBL) system with transducers placed at four corners of the ASV (or even wider) is a better choice than an ultra-short baseline (USBL) system. SBL systems use trilateration of range measurements from the bottomside unit relative to the topside transponders to determine the relative position of the underwater vehicle relative to the surface vehicle. With a larger baseline, the SBL underwater localization system is less sensitive to range measurement noise.

It is omidirectional and provides the ability to change both the expected maximum range and the bearing angle sector to improve positioning accuracy. The UWGPS G2 system is equipped with an on-board computer that takes GNSS, IMU and acoustic measurements and converts them to the ROV’s relative position with respect to the ASV and to georeference coordinates. There is also the capability to provide the topside with external GNSS and IMU measurements, either via NMEA messages or as HTTP requests to the local HTTP servers that perform all calculations and communications. The ping rate is 2–4 Hz, which is more than sufficient for the ROV with a maximum forward speed of 3 kt.

It offers Ethernet and WiFi communication interfaces. Ethernet is more suitable for autonomous HEKTOR missions because it has greater bandwidth and less interference than WiFi, and does not interfere with the WiFi over which the vehicles communicate. Setting of all parameters of the system is facilitated by a web GUI that runs on the topside computer and is accessible through any web browser. This web GUI also allows the user to track the ROV with its estimated global coordinates on a georeferenced map, as shown in Figure 13b.

During initial testing at the Breaking the Surface marine robotics workshop in October 2021 in Biograd na Moru, Croatia, the transponders were attached to metal rods and mounted on thruster mounts at the front and on the rear wheel axles at the rear. They are mounted below the thrusters to avoid acoustic noise. They are also mounted at various distances (15, 25, 35 and 45 cm below the thrusters, i.e., at 50, 60, 70 and 80 cm depths) to increase positioning accuracy, as recommended by the manufacturer. The baseline in this configuration was 1.46 × 0.487 m.

Locator U1 (see Figure 14b) is a lightweight (75 g in water) battery-powered bottomside transponder. Its battery lasts up to 10 h and is easily recharged via a USB-C port (∼6) h. It is rated for up to 300 m. It has an integrated GNSS chip for time synchronization with the topside unit. The bottomside unit mounts to the Blueye ROV with a bracket sold by WaterLinked. The frequency channel is selected by a switch in the cap of the Locator U1, as shown in Figure 14b.

On the software side, it is easy to integrate the system software-wise thanks to the Python v3.9 API developed by the manufacturer. An ROS2 package was developed to integrate the acoustic localization measurements into the ROS framework. It sends HTTP requests to the topside HTTP server, parses the returned relative and global position and publishes them in ROS Topics. The code of the package is open-source and publicly available at [52].

Normally, after a successful IMU calibration (as shown in Figure 14c), the topside computer can accurately estimate the ROV’s relative position with respect to the topside based on trilateration and the topside’s known GNSS and heading. However, due to the erroneous IMU, the calibration was unsuccessful during the trials in Biograd na Moru, Croatia in October 2021. This meant that alternative corrections had to be made. The first solution was to set the GNSS and IMU sources as static, i.e., a fixed GNSS position near the test site with heading 0° to the north, as shown in Figure 13b. In this configuration, the ROV’s relative positions were not corrupted, but its transformed global position estimates were. This problem was solved in the post-processing phase by fusing the relative ROV positions with the ASV position in the local NED frame and its 3-degrees-of-freedom orientation. Post-processing results are shown in Figure 14d.

### 5.3. Tether Management System and ROV Docking Mechanism

The integration of the ROV on the ASV and their cooperative path planning during autonomous inspection missions requires the development of a tether management system (TMS) together with a docking mechanism (DM) for the ROV, as shown in Figure 15. The 400 m long, specially cut tether, which was purchased along with the 300 m rated ROV, must first be wound onto a drum. When the autonomous fish net pen inspection mission begins, the docking mechanism opens and the ROV dives to inspect the fish net pens. During the mission, a stepper motor connected to the tether drum via a shaft controls the length of the unwound ROV tether and ensures that the TMS does not interfere with the ROV’s movement. The underwater acoustic localization system estimates the position of the ROV relative to the ASV to help decide whether to wind or unwind the tether. Once the ROV is finished inspecting the net, it surfaces and moves closer to the ASV. During this time, the TMS also reels up the tether and pulls the ROV closer. Once the ROV is between the catamaran’s hulls, the V-shaped docking mechanism closes, securing the ROV in place.

The control system of the TMS consists of a 2 Nm, 5 A ( IP65 rated) Nema 23 bipolar stepper motor with a built-in encoder and a CL57T-V4.0 closed-loop stepper motor drive with an output current range of ${I_{driver}^{TMS}}^{out}$ = 0–8.0 A and an input voltage range of ${U_{driver}^{TMS}}^{in}$ = 24–48 VDC. To increase the motor torque, the microstepping factor was set to $K_{\mu step}$ = 1, so that the motor would make its rated steps per revolution $SPR=SPR^{n}K_{\mu step}$ = 200.

Since the maximum output voltage of the DC /DC converter to which the stepper motor driver is connected is $u_{i}$ = 35 V and not 48 V, this means that the motor can achieve a maximum of 3.66 A to its phases. With a maximum input current of $I_{m}^{max}$ = 3.66 A and a maximum rated speed of $RPM_{max}^{n}=$ 1500 rpm, this means that the motor can rotate at most ${rpm}_{max}=$ 686.4 rpm at approximately 1.16 Nm.

In the development phase of the prototype, all electronics of the TMS are housed in an IP66-rated plastic “control box”. This includes the stepper motor driver, the associated DC /DC converter, an Arduino Uno with its 10 V voltage stabilizer and a BlueRobotics 14.8 V, 18 Ah battery. The box has two cable glands on the side, one for the external power source and USB cable to communicate with the main computer on board the ASV, and one for the cable to power the motor. The power button, motor rotation direction switch and motor speed control potentiometer, all used in the TMS manual control mode, are well sealed to prevent water from entering the control box. To test the performance of the TMS without having to use the ASV, a prototype was developed for use in the laboratory pool, as shown in Figure 15b. It is mounted with the main stepper motor of the TMS, connected to the control box, and was successfully tested.

### 5.4. ROS Autonomy and Collaboration Framework

Nowadays, ROS is omnipresent in robotics research and application. It is an open-source middleware that enables software development of distributed processes responsible for specific tasks. ROS enables easy communication between processes via Topics, where each process is represented as a node in a graph. Messaging in ROS1 is not inherently real-time, but this problem has been addressed in the ROS2 framework. Communication of heterogeneous hardware and software is also simplified through the use of ROS so that control of devices and processes across the spectrum from low-level device control to high-level navigation, control, path and mission planning algorithms can be integrated into a unifying framework. In addition, communication of modules running on different ROS1 versions and even ROS1–ROS2 message passing is possible. This makes the integration of different vehicles possible and enables complex systems and the cooperative behavior of autonomous vehicles without having to worry about low-level integration. The integration of the HEKTOR system via an ROS middleware is shown in Figure 16.

ATMM and LAAR are used in the viticulture scenarios. Both are equipped with Intel NUC mini-PCs running Ubuntu with ROS. The ROS packages on ATMM communicate via CAN with track motor drives used for ground movement. This is then coordinated with the ROS packages responsible for controlling the Kinova robotic arm for protective spraying and suckering. These operations rely on local sensor information collected by ROS sensor packages, including camera, LiDAR, IMU, GNSS and others. Additional sensor information is received from LAAR via secure WiFi, which is achieved through a multi-master ROS scheme. Based on the information collected by the sensors and received by LAAR, ATMM’s ROS autonomy packages generate references and send them to the track motor drives and Kinova robotic arm to autonomously perform the required tasks in the vineyard.

Low-level control of the electric motors on LAAR is implemented on the Pixhawk autopilot module, which communicates via a MAVROS package with the NUC mini-PC running ROS. This allows the NUC to issue flight commands via ROS and receive flight telemetry that includes GNSS information, IMU data, air pressure, battery and motor status. Other ROS packages on the NUC enable communication with various sensors, which primarily include various cameras and LiDAR. The ROS path generation package will generate a path with lawnmower patterns for mapping and monitoring vineyards based on the predefined vineyard boundaries. Based on the path and current sensor measurements, the ROS autonomy package will generate references for the autopilot module to follow the desired path. The multi-master ROS scheme allows the exchange of ROS Topics via WiFi with ATMM, which include GNSS coordinates and a generated map of the vineyard.

As mentioned earlier, there will be three types of vehicles in mariculture scenarios, namely LAAR, ASV and ROV. The main computers onboard the LAAR and ASV both run at ROS. The Blueye ROV runs the user-upgradeable UNIX-like Blunux operating system, but does not use ROS, although the structure of Blunux processes is similar. The LAAR uses ROS1 Melodic, while the ASV uses both ROS1 and ROS2. The navigation, guidance and control (NGC) modules of the ASV are executed in ROS1 Noetic to ensure compatibility with previously developed software used on similar ASVs developed at the Laboratory of Underwater Systems and Technologies (LABUST) at the University of Zagreb. All new features added to the ASV in the HEKTOR project are mostly implemented in ROS2 Foxy. This includes the interface for communication and control of the ROV, as mentioned in Section 4.3, processing of its visual data, but also the streaming and processing of all new sensors integrated into the ASV or to be integrated in the future, as mentioned, for example, in Section 5.2. This was the first step to fully port all modules to ROS2.

Communication between the ROV and the ASV is enabled in ROS2 by the Blueye SDK-RO2 interface, which was developed as part of the HEKTOR project and is available as open source at [46]. As mentioned in Section 4.3, this interface streams video from the ROV camera to the ASV onboard computer on the topside, while all commands from the topside are sent to the ROV via the ROS2-Blueye SDK interface. In this way, the ROV is autonomous in a sense. The interaction between the ASV and the ROV is defined by the type of mission that the vehicles need to perform. The ASV and ROV communicate via the tether, but the interaction depends on the position of the ROV relative to the ASV and the fish net pens. The ASV is capable of locating the ROV using the acoustic localization system built into ROS2, as described in more detail in Section 5.2. In addition, as described in Section 5.3, the ASV has a TMS subsystem that, depending on the distance of the ROV from the ASV (provided by the acoustic localization ROS module), controls the winding and unwinding operations of the tether management system via ROS.

Cooperation between the ASV and the LAAR is based on the transmission of latitude and longitude coordinates of the two vehicles for long-range missions via a secure WiFi network. All this can be achieved through a multi-master ROS scheme. For security reasons, each vehicle should define publicly accessible topics so as not to leak all internal states and measurements to the outside world. The ASV should also provide the LAAR with its IMU measurements, which are useful for the LAAR to optimize its autonomous landing motion planning or to temporarily abort the landing if the ASV rolls or pitches too much due to waves. In addition, the LAAR should notify the ASV as soon as it lands on the landing platform and activates the landing skis’ electromagnets. This signals the ASV to close the landing platform, which is described in detail in Section 5.1. Successful closing of the LP guides signals the LAAR to turn off its electromagnets.

## 6. Conclusions and Future Work

This paper presents the development of robotic vehicles within the HEKTOR project, together with their integration into the system used for autonomously executed tasks in viticulture and mariculture.

The viticulture scenarios include tasks such as vineyard monitoring, spraying and suckering based on coordinated actions of the mobile manipulator and the UAV, while the mariculture scenarios include tasks such as inspecting net pens from below, from the water surface and from the air, as well as modeling fish populations in net pens. The challenges and state of the art for each of these tasks are presented, along with their solution within the HEKTOR project, which is based on four different robots.

The specifications and characteristics of the all-terrain mobile manipulator (ATMM), the aerial robot (LAAR), the autonomous surface robot (ASV) and the underwater robot (ROV) are presented, along with the subsystems used for collaboration between them. These include the landing platform, the underwater localization system, the underwater tether management system and the ROV docking mechanism. The ROS-based framework used by all four robots enables the modularity and autonomy of each robot, as well as the possibility of cooperative actions with two or more different robots.

Future work includes the implementation of autonomous behavior and field testing of the HEKTOR system. In viticulture scenarios, this will include the autonomous monitoring of vineyards and generation of maps by LAAR, and ATMM navigation based on these maps to perform spraying and suckering tasks. In mariculture scenarios, LAAR autonomous landing will be implemented on the ASV landing platform. In addition, autonomous inspection of net pens will be performed both from the air (LAAR) and underwater (ROV) based on the implemented UWGPS and tether management system.

## Figures and Tables

**Figure 1 sensors-22-02961-f001:**
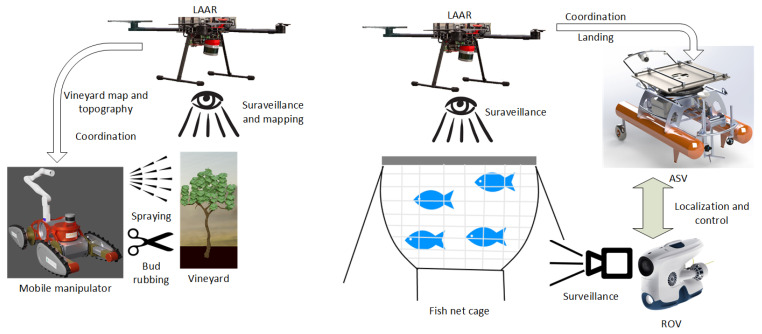
HEKTOR system overview. Viticulture scenarios: vineyard surveillance, protective spraying and suckering with an all-terrain mobile manipulator (ATMM) and a light autonomous aerial robot (LAAR). Mariculture scenarios: coordinated monitoring of fish net pens from below and from the air using the LAAR, an autonomous surface vehicle (ASV), and a remotely operated underwater vehicle (ROV).

**Figure 2 sensors-22-02961-f002:**
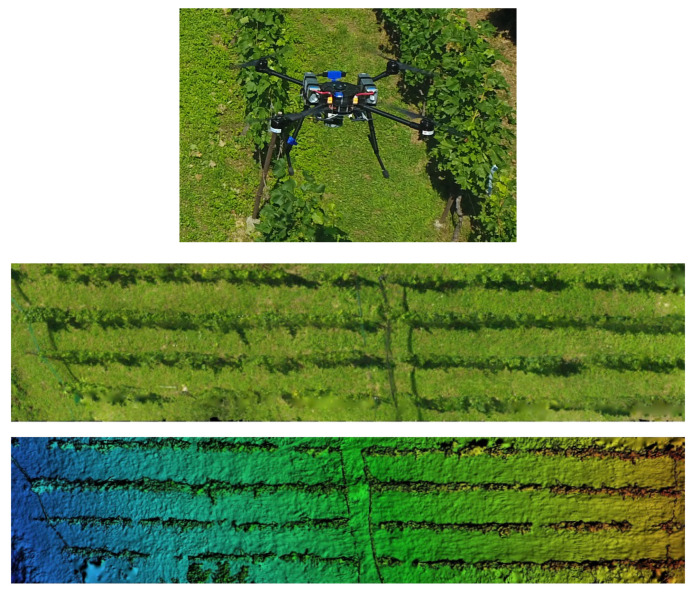
Monitoring the vineyard from the air, **top**—LAAR monitoring the vineyard, **middle**—stitched RGB image, **bottom**—generated 3D model.

**Figure 3 sensors-22-02961-f003:**
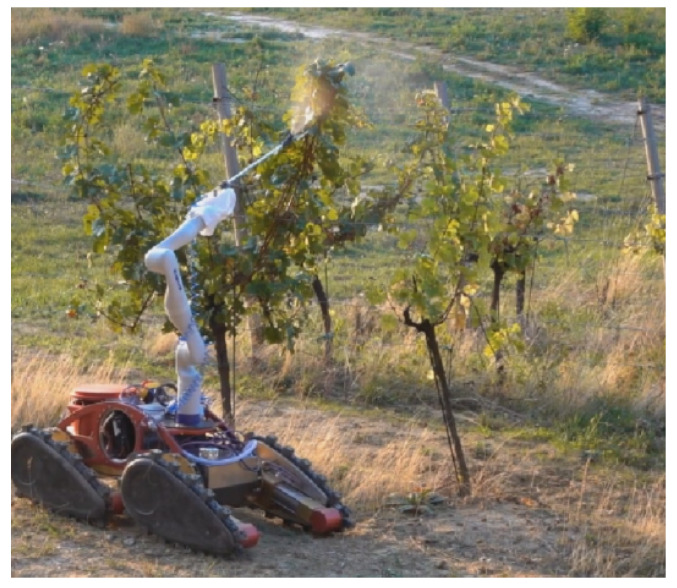
An ATMM in a vineyard spraying scenario. Spraying nozzle is attached to the robot arm, and the arm motion controls its pose relative to the mobile base.

**Figure 4 sensors-22-02961-f004:**
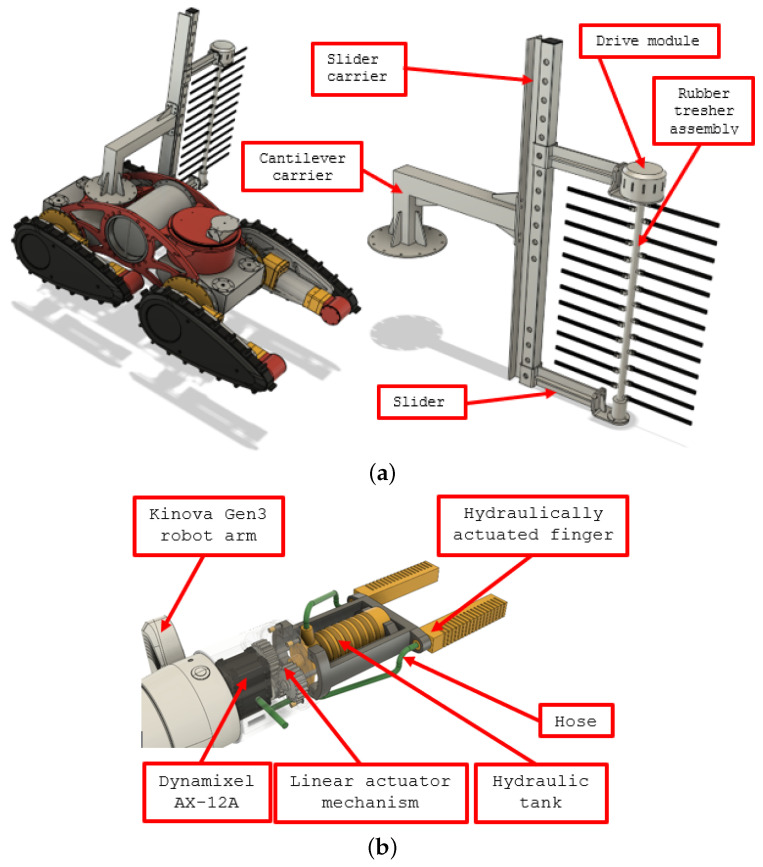
Suckering specialized tools: (**a**) standalone rubber threshing system, (**b**) compliant hydraulically actuated gripper.

**Figure 5 sensors-22-02961-f005:**
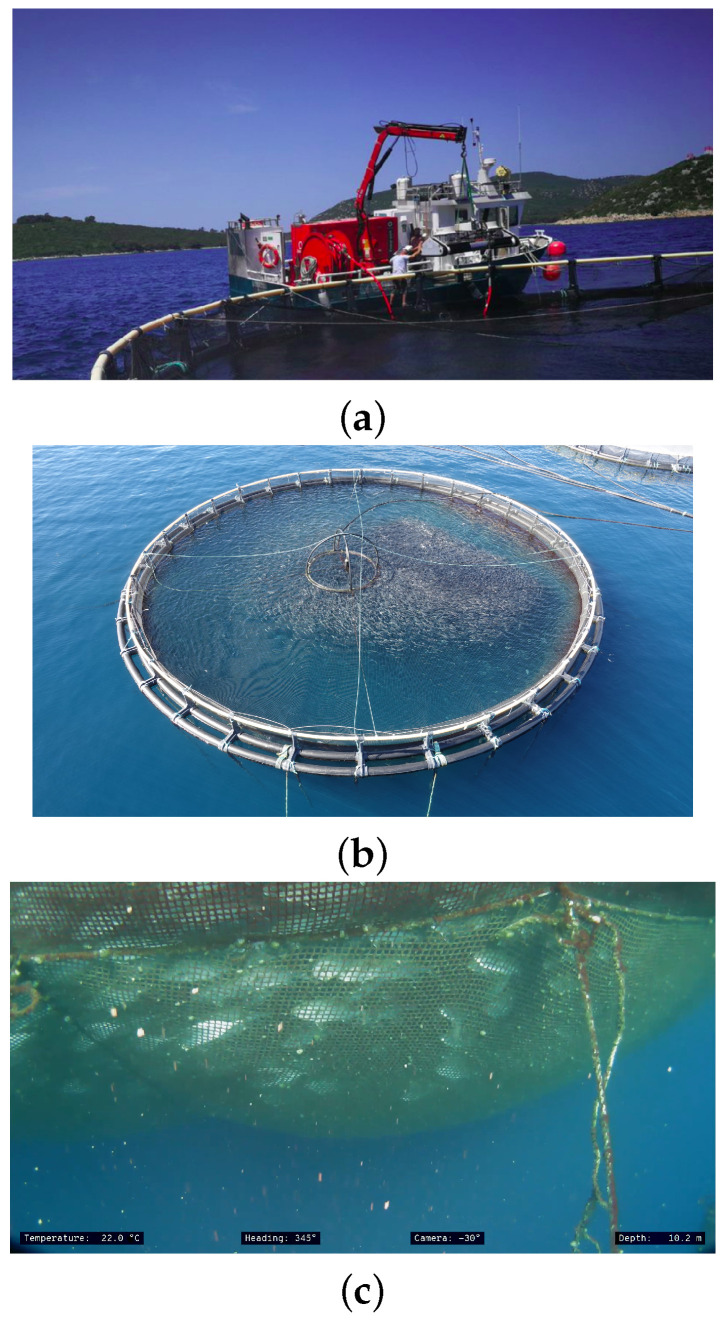
(**a**) Commercial, manually controlled machine for undersea net cleaning. (**b**) Image taken from inspection LAAR flying above the net pen. (**c**) Undersea images of net pens taken by an inspection ROV.

**Figure 6 sensors-22-02961-f006:**
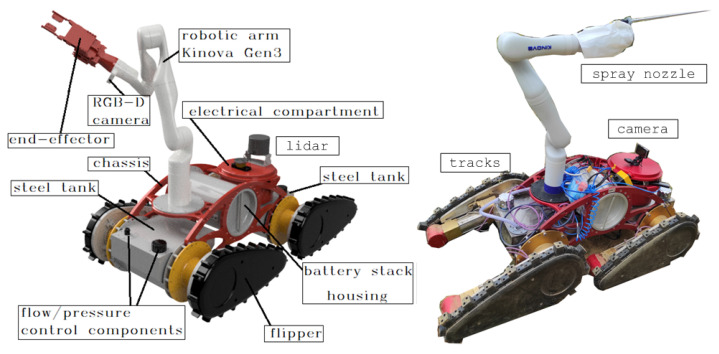
Design of an ATMM consisting of a mobile platform driven by four independently controlled flippers/tracks and a 7-degrees-of-freedom (DoF) robot arm: (**left**) equipped with a gripper for suckering, (**right**) equipped with a spray nozzle for controlled spraying.

**Figure 7 sensors-22-02961-f007:**
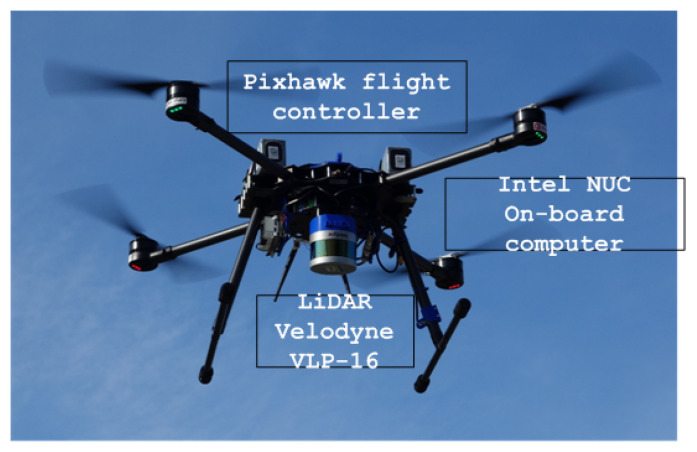
LAAR—carbon-based quadcopter with Pixhawk flight controller, Intel NUC on-board computer and Velodyne VLP-16 LiDAR sensor.

**Figure 8 sensors-22-02961-f008:**
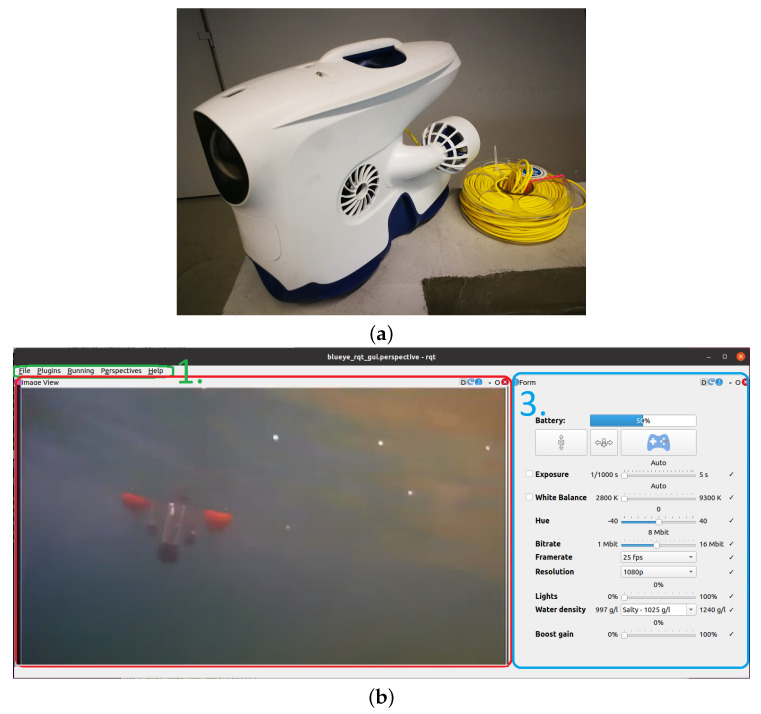
(**a**) Blueye Pro ROV. (**b**) Graphical user interface of Blueye-ROS2 interface.

**Figure 9 sensors-22-02961-f009:**
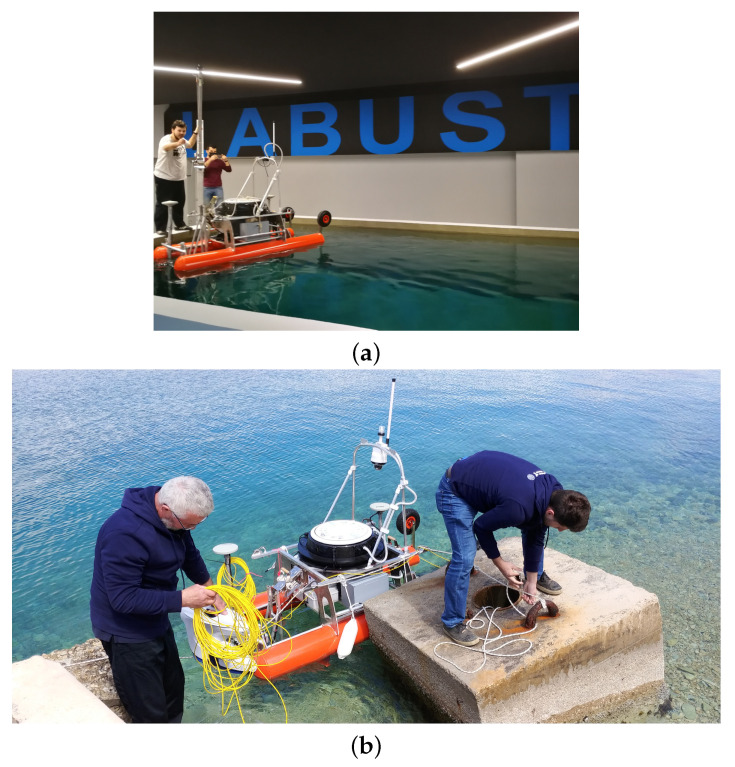
(**a**) Testing of the ASV Korkyra at LABUST’s pool. (**b**) Maiden voyage and testing of ASV and ROV controls system integration at sea in Split, Croatia in April 2021. (**c**) Ouster OS1-128 lidar and FLIR Blackfly S USB camera integration testing at sea in Biograd na Moru, Croatia in October 2021.

**Figure 10 sensors-22-02961-f010:**
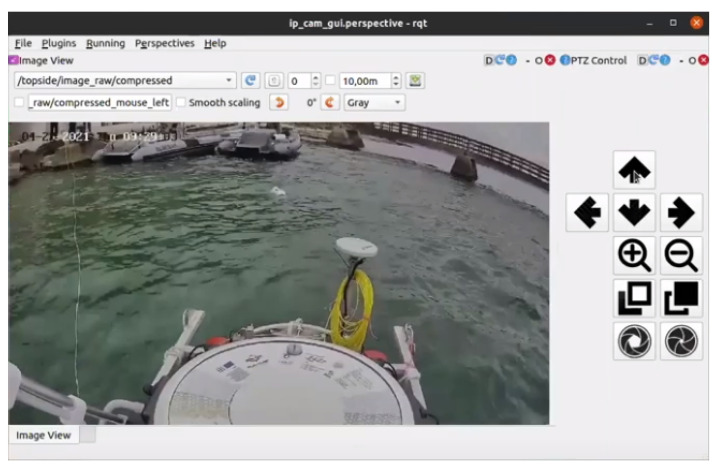
User interface of the catamaran’s surveillance pan-tilt-zoom camera. System integration tests in Split, Croatia in April 2021. Blueye ROV is visible just in front of the ASV.

**Figure 11 sensors-22-02961-f011:**
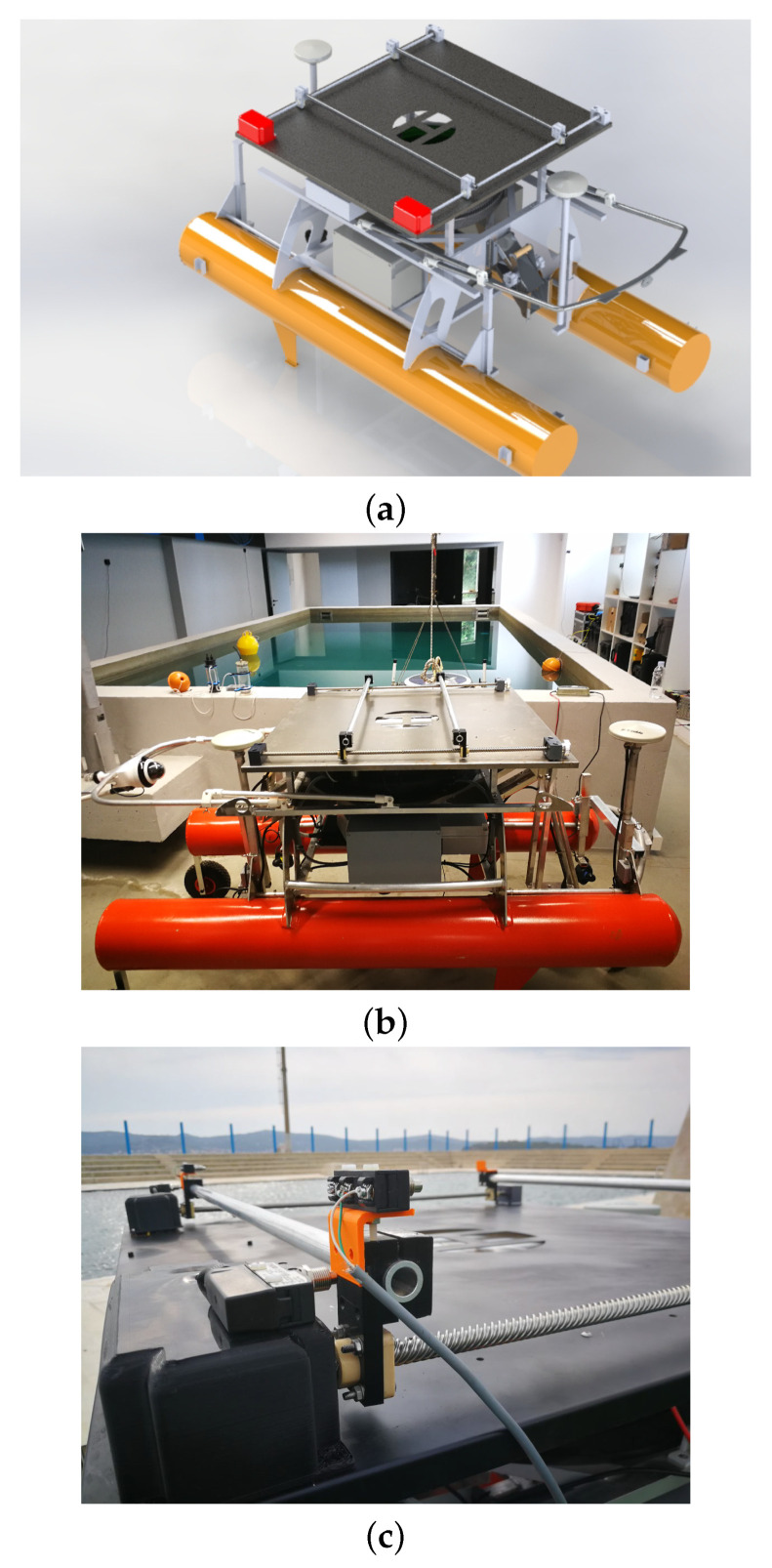
Platform for landing the LAAR on ASV Korkyra. (**a**) SolidWorks model rendering. (**b**) Constructed prototype version for laboratory testing, mounted on the ASV. (**c**) Stepper motors (in black boxes) and four integrated contact switches for the control system of the LP.

**Figure 12 sensors-22-02961-f012:**
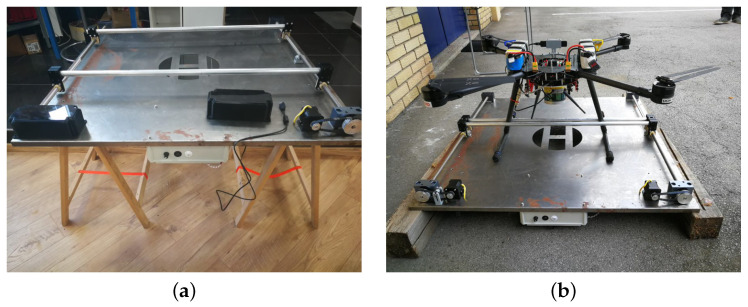
Test setups for the landing platform. (**a**) Test setup in the laboratory with a control box. (**b**) Outdoor test setup with manual landing of the LAAR and semi-automatic control of the LP in Zagreb, Croatia.

**Figure 13 sensors-22-02961-f013:**
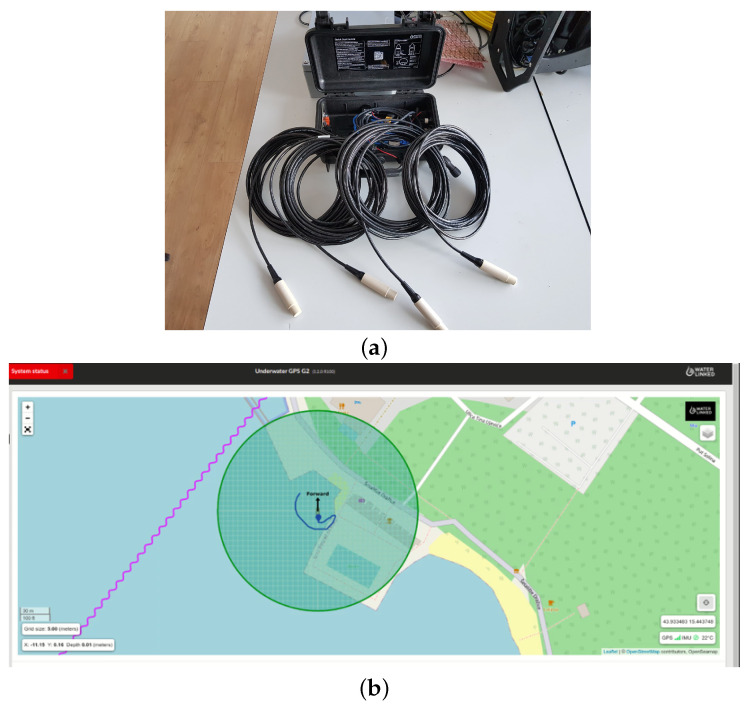
(**a**) WaterLinked Underwater GPS G2 topside with 4 transducers for underwater acoustic localization. (**b**) Web GUI on the topside computer that allows system setup and visual tracking of the ROV on a georeferenced map.

**Figure 14 sensors-22-02961-f014:**
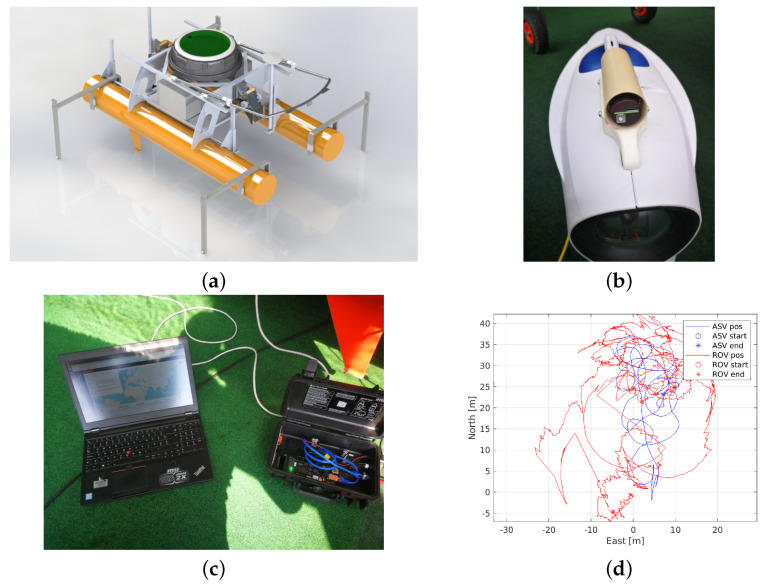
(**a**) The U1 locator is mounted on a bracket on top of the Blueye Pro ROV. The cap is open to show the USB-C charging port and the rotary switch for channel selection. (**b**) Calibration of the topside IMU. (**c**) Relative positions of the ROV merged with the ASV heading in the local NED frame. (**d**) Georeferenced positions of the ROV.

**Figure 15 sensors-22-02961-f015:**
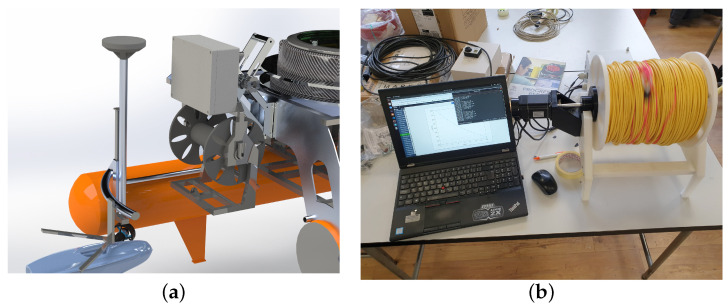
(**a**) SolidWorks model of the tether management system and docking station for the ROV aboard the ASV Korkyra. Side view of the assembly on the ASV. (**b**) Laboratory testing of the prototype TMS.

**Figure 16 sensors-22-02961-f016:**
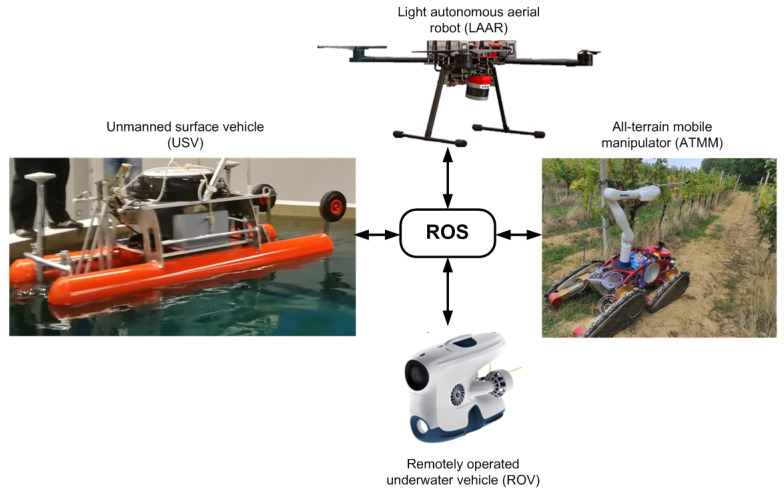
HEKTOR system overview. All software development is based on the use of ROS.

**Table 1 sensors-22-02961-t001:** Overview of current approaches for vineyard surveillance, spraying and suckering compared to the developed HEKTOR system.

Ref.	Tasks in Viticulture			Vehicles		Unstruct. Env.
	Surveillance	Spraying	Suckering	LAAR	ATMM	
[5]	X					X
[6]	X				X	X
[7]	X			X		X
[8]	X			X		X
[9]	X			X		X
[10]	X			X		X
[11]	X			X		X
[12]	X			X		X
[13]	X			X		X
[14,15]		X			X	
[16,17]		X			X	
[18]		X	X			X
[19]			X			X
HEKTOR	X	X	X	X	X	X

**Table 2 sensors-22-02961-t002:** Overview of current approaches for net pen inspection and biomass estimation compared to the developed HEKTOR system.

Ref	Tasks in Mariculture			Vehicles			Unstruct. Env.
	Net Pen Inspection	Biomass Estimation	Feeding	LAAR	ASV	ROV	
[27]			X	X			X
[28]			X	X			X
[29]	X			X			X
[30]	X			X			
[31]			X	X			
[32]	X					X	X
[33]	X					X	X
[34]	X					X	X
[35]	X				X	X	X
[36]		X					
[37]		X					
[38]		X					
[39]		X					
[40]		X					
[41]		X					
[42]		X					
[43]		X					
[44]		X		X	(X)	X	X
HEKTOR	X	X		X	X	X	X

**Table 3 sensors-22-02961-t003:** The list of selected HEKTOR system requirements.

HEKTOR System Requirements for Viticulture Scenarios	
Terrain characteristics	ATMM should function on slopes from 0% to 60%, on rocky, earthy or seismic terrain.
Vineyard size	The estimated vineyard size for application of HEKTOR system without human intervention is 1 ha, with an average planting height of 1.5 m and row spacing of 1.2–2 m.
Permissible flying height	Aerial survey of vineyards should be carried out at a height of at least 10 m above the height of the plantation. When mapping vineyards, the maximum height depends on the area of the vineyard (≤30 m).
Permissible wind state	UAV should be able to fly and execute its tasks at wind speeds up to 15 m/s.
3D map of vineyards	The HEKTOR system should produce a 3D map of vineyards with an accuracy of 10 cm.
Reliable communication	The operator’s system should communicate reliably with the HEKTOR system at a distance of 150 m.
Localization ATMM	ATMM should know its position in space with an accuracy of 10 cm.
Spraying efficiency	ATMM should treat plantations with a travel speed of at least 0.7 m/s at a slope of up to 30%.
Suckering efficiency	ATMM shall achieve a suckering rate of at least 20 vines per hour.
HEKTOR System Requirements for Mariculture Scenarios	
Operating depth ROV	ROV should work at depths up to 300 m.
Weather conditions	ROV should work in fish farms at currents up to 2 knots.
ROV maneuverability	ROV should be able to move in 4 degrees of freedom, namely: yaw (rotation around the z-axis, i.e., rotation left–right), sinking (movement along the z-axis, i.e., movement up and down), heading (movement along the x-axis, i.e., forward–backward movement), and drift (movement along the y-axis, i.e., lateral left–right movement).
Localization ROV	ROV should know its position in space with an accuracy better than 1 m.
Weather conditions LAAR	The LAAR must be able to fly safely with the wind speed up to 15 m/s.
Weather conditions ASV	The ASV must be able to successfully complete missions at sea state 2.
ASV maneuverability	ASV should be able to move in 3 degrees of freedom, namely: yaw (rotation around the z-axis, i.e., rotation left–right), heading (movement along the x-axis, i.e., movement back and forth), and drift (movement along the y-axis, i.e., lateral movement left–right).
Reliable communication	The operator’s system should communicate reliably with ASV at a distance of 150 m.
Localization ASV	ASV should know its position in space with an accuracy better than 1 m.

**Table 4 sensors-22-02961-t004:** Robots of the HEKTOR system.

Robot	ATMM	LAAR	ROV	ASV
**Type**	Terrestrial	Aerial	Underwater	Water surface
**Dimensions**	709 × 565 × 1327	1200 × 1200 × 450	485 × 257 × 354	2000 × 1000 × 1400
**W × L × H [mm]**				
**Weight [kg]**	90 + 10	8	9	100
**Payload [kg]**	>200	2	n/a	100
**Battery [Wh]**	1248	2 × 266	96	3730
**Autonomy [min]**	300	30	120	600
**Actuators**	4 × 250 W +	4 × 1600 W	4 × 350 W	4 × 390 W +
	7DoF arm 36 W			1 × 720 W
**Speed**	0.7 m/s	>10 m/s	3 kt	4 kt
**Basic**	Camera	Camera	Camera	Camera
**sensors**	LiDAR	LiDAR	IMU	IMU
	IMU	IMU	Pressure	GNSS
	GNSS	GNSS	Temperature	
**Optional**	Flow	Thermal camera		Multibeam sonar
**sensors**	Pressure	Multispectral camera		LiDAR
**Communication**	WiFi + Radio	WiFi + Radio	Ethernet + WiFi	WiFi + Radio + LTE
**Control unit**	NUC 10	NUC 11	N/A	NUC 7 +
	+ Pixhawk	+ Pixhawk		NUC 10
	CubeOrange	CubeBlack

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
