# Peer review of "Heterogeneous Autonomous Robotic System in Viticulture and Mariculture: Vehicles Development and Systems Integration†"

_sensors, 2022, doi:10.3390/s22082961_

Round 1
Reviewer 1 Report
This paper presents the robotic components of a heterogeneous autonomous robotic system in viticulture and mariculture.
The authors did not indicate what the contributions of this paper clearly.
There is a lack of technical subsistence. The novelty of the article is not compared to the state-of-the-art sufficiently in the introduction section.
It is essential to demonstrate the problem at stake and the drawbacks of existing studies when proposing the approach.
Reviewer 2 Report
This work presents the robotic components of a heterogeneous autonomous robotic system, divided into two main parts dealing respectively with viticulture and mariculture. In viticulture, this includes vineyard surveillance, spraying and suckering with an all-terrain mobile manipulator and a light autonomous aerial robot. In mariculture, scenarios include coordinated aerial and subsurface monitoring of fish net pens. All robotic components communicate and coordinate their actions through the Robot Operating System. Capabilities of the system are demonstrated by field tests.
The topic is original and relevant in the field, this work presents and integration of systems whit a high potential for developing collaborative robotics in challenging environments such as viticulture and mariculture. This paper presents the development of the HEKTOR project, together with their integration into the system used for autonomously executed tasks in viticulture and mariculture.
The conclusions are supported by the results and the references are appropriate. The provided information is relevant for the knowledge field. This manuscript could be considered for publication.
Reviewer 3 Report
This paper presented the development of robotic vehicles within the HEKTOR project, together with their integration into the system used for autonomously executed tasks in viticulture and mariculture. The whole system is a practical system and it would be of interest to readers. However, the present description and introduction of the system is a bit two superficial and it lacks the original contribution. So major revisions are needed:
- Page 2 The last paragraph for the introduction of the project HEKTOR, the jump of figure number from 1 to 5 is a bit strange. Actually these two figures can be merged into one for the complete HEKTOR system, LAAR is common, in Fig.1, add another fish net, USV and ROV in Fig 1. This could save the space and make more clear of the advantage of the present concept. This figure is moved in Introduction section. It also matches with Fig.17.
- Page 2 line 72, “the occurrence of diseases and the yield of grapes. In [5], the authors estimate”, Here add one sentence to describe what main tasks are done and then introduce others work in solving these tasks.
- Page 2 in the first paragraph of section 2.1, What are the advantages and disadvantages of these existing systems and what improvements are attempted in HEKTOR?
- Page 4 lines 113-116, “The use of robots for vineyard spraying opens up a number of opportunities to reduce the amount of chemicals introduced into the environment by using artificial intelligence to optimize the chemicals used to the minimum necessary”. The logic here needs more explanation. To spray chemical agents may harm the operators and use Robots to reduce this harmful work. However, whether to reduce the amount of chemicals depends on the availability of other protective non-chemical products which has nothing to do with the use of robots?
- Page 5 line 160, “Regardless of which path to solving this task would be faster and more efficient, the development of suckering…”, this sentence is hard to be understood, please modify.
- In section 2 last paragraph, Please explain in a bit more detail about these two design solutions. This is the actual original contribution of the present work. At present the description of your own design is too superficial.
- Page 5 lines 167-168, Fig.5 is merged with Fig.1 and also Please unify to use either USV or ASV but not in a mixed way of confusing.
- Page 7 before line 207, The review to the existing work is superficial and the advantages and disadvantages of each system must be analyzed and then lead to the main consideration or requirement for the HEKTOR system.
- Page 8 lines 248-249, “In the HEKTOR project, information collected both below the sea surface and from the air is analyzed for fish population modeling”. What is the necessity and advantage to use the information from the air?
- Page 10, in the label of Fig.7, The explanation to the left and right of two figures should be provided.
- In Table 1 of Page 11, At what wind state, LAAR can fly? At what sea state ASV and ROV can operate? Please specify the complete working condition. For ASV, it is mentioned sea state 2, this is very low, How many days can work for a recent statistics in this region?
- Page 12, for the introduction of LAAR, At what wind speed it can flight?
- Some minor spelling and format mistakes are marked in the attached PDF.

Reviewer 4 Report
This paper focuses on the development of two types of robotic components of heterogeneous autonomous robotic systems for the HEKTOR project, which can be used to handle viticulture and mariculture, respectively. The tests show that the HEKTOR project can be used in several scenarios and has a strong capability to guarantee the quality of the executed work while meeting the quantitative necessities. However, there are still some minor problems in the paper.
- the number of tables in the paper is too small, the authors could have illustrated the final results with some graphical data to highlight more intuitively the effectiveness of the proposed HEKTOR system.
- The format of the references needs to be standardized.
Round 2
Reviewer 1 Report
I have no further question.
Reviewer 3 Report
I am happy with the revised version.